# Stress Shielding and Bone Resorption of Press-Fit Polyether–Ether–Ketone (PEEK) Hip Prosthesis: A Sawbone Model Study

**DOI:** 10.3390/polym14214600

**Published:** 2022-10-29

**Authors:** Seyed Ataollah Naghavi, Churun Lin, Changning Sun, Maryam Tamaddon, Mariam Basiouny, Pilar Garcia-Souto, Stephen Taylor, Jia Hua, Dichen Li, Ling Wang, Chaozong Liu

**Affiliations:** 1Institute of Orthopaedics & Musculoskeletal Science, Division of Surgery & Interventional Science, University College London, Royal National Orthopaedic Hospital, Stanmore, London HA7 4LP, UK; 2Department of Mechanical Engineering, University College London, London WC1E 7JE, UK; 3State Key Laboratory for Manufacturing Systems Engineering, School of Mechanical Engineering, Xi’an Jiaotong University, Xi’an 710049, China; 4National Medical Products Administration (NMPA), Key Laboratory for Research and Evaluation of Additive Manufacturing Medical Devices, Xi’an Jiaotong University, Xi’an 710054, China; 5Medical Physics & Biomedical Engineering, University College London, London WC1E 6BT, UK; 6School of Science and Technology, Middlesex University, London NW4 4BT, UK

**Keywords:** additive manufacturing, biomechanical behaviors, bone resorption, finite element analysis, hip implant, hip stiffness, orthopaedic implants, PEEK, polyetheretherketone, stress shielding

## Abstract

Stress shielding secondary to bone resorption is one of the main causes of aseptic loosening, which limits the lifespan of the hip prostheses and increases the rates of revision surgery. This study proposes a low stiffness polyether–ether–ketone (PEEK) hip prostheses, produced by fused deposition modelling to minimize the stress difference after the hip replacement. The stress shielding effect and the potential bone resorption of the PEEK implant was investigated through both experimental tests and FE simulation. A generic Ti6Al4V implant was incorporated in this study to allow fair comparison as control group. Attributed to the low stiffness, the proposed PEEK implant showed a more natural stress distribution, less stress shielding (by 104%), and loss in bone mass (by 72%) compared with the Ti6Al4V implant. The stiffness of the Ti6Al4V and the PEEK implant were measured through compression tests to be 2.76 kN/mm and 0.276 kN/mm. The factor of safety for the PEEK implant in both static and dynamic loading scenarios were obtained through simulation. Most of the regions in the PEEK implant were tested to be safe (FoS larger than 1) in terms of representing daily activities (2300 N), while the medial neck and distal restriction point of the implant attracts large von Mises stress 82 MPa and 76 MPa, respectively, and, thus, may possibly fail during intensive activities by yield and fatigue. Overall, considering the reduction in stress shielding and bone resorption in cortical bone, PEEK could be a promising material for the patient–specific femoral implants.

## 1. Introduction

Total hip arthroplasty (THA), also known as total hip replacement (THR), is refers to the replacement of the diseased hip joint with an artificial hip prosthesis to restore the patient’s mobility and improve the quality of life. THA is a clinically successful operation, which has been developed and applied in clinical practice for more than 3 decades; however, recent projections indicate an increase in the number of revision surgeries by almost 137 percent in the next 15 years [1]. Revision surgeries are generally required when the implants are seriously damaged or loosened, which can be painful and costly to the patients. Although most hip implants are designed to last for at least 20 years, the life span of these implants is still far from satisfactory as more and more young patients are involved in THR and lower success rates are generally reported in long–term use [2]. Hence, an extension in the life span of hip implants and a reduction in the rates of revision surgeries is equally of a significance.

One of the major post-surgery complications is aseptic loosening. It is induced by four mechanisms: mechanical failure of the implant or cement, the introduction of wear debris into the interface region causing osteolysis, relative motion across the interface, and stress shielding in the bone [3,4]. Mutable biological responses can be initiated through these mechanisms, which lead to bone resorption and eventual loosening of the implant. Noticeably, stress shielding is the main cause of aseptic loosening in long-term service, which is commonly revealed in clinical cases [5]. It is caused by insufficient load transfer between bone and the implants. Based on Wolff’s law, bones adapt to the mechanical load they receive, suggesting that when a person is much more active in a particular region of the body, more bone is produced to strengthen the tissue, and when a bone is unloaded for an extended period, the tissue mass can diminish accompanied by bone resorption. In post-surgery conditions, only part of the load is transferred to the bone, while most of the load is carried by the implant [6]. This altered stress profile is expected to cause mineral bone lost in the interface of bone and implants leading to loosening due to the lack of contact in the interface.

The stress distribution in the femur is largely affected by the mechanical properties of the hip prosthesis. In a composite component, materials with higher stiffness are expected to attract larger stress compared with less stiff materials. Therefore, a large mismatch in stiffness between natural bone and the implant can lead to a severe stress shielding effect [7]. In current THA design, metals, such as cobalt–chromium alloys, 316L stainless steel, and titanium and its alloys (e.g., Ti6Al4V, Ti–Mg), are the predominant materials of choice for femoral stems [8,9,10]. These materials have noticeably high stiffness compared to the femur. For instance, Ti and its alloys are widely applied in THA with the advantages of high biocompatibility and high fatigue strength. Although they have relatively lower Young’s modules (110 GPa) than other metals, it still remains much higher than that of natural bone (1–30 GPa) [11]. 

Previous research tackled stress shielding either by geometric design, lattice structures or applications of novel materials. For geometric design, Kuiper and Huiskes proposed a femoral stem with a non–homogeneous distributed Young’s modulus by numerical design optimization methods. Compared to conventional stems with a homogeneous low modulus, the interface stress of their design was reduced by more than 50% [12]. Hanada et al. [13] developed a novel new fabrication method producing a cementless stem with a gradient Young’s modulus (~40 GPa), which utilizes the newly designed β–type Ti–33.6Nb–4Sn alloy (TNS stem). A proximal load–transfer pattern and good initial stability of the TNS stem is claimed through in-vitro biomechanical study. Similarly, Sun, C. et al. [14] developed algorithms to adjust the elastic modulus of Functionally Graded Material in porous hip stem design. A 40% bone loss reduction compared to the solid stem was obtained through FE simulation. Gross and Abel designed a hip implant with a hollow structure and evaluated its advantages in the reduction in stress shielding (15–32%) by finite element analysis [15]. Tan et al. [16] studied two topology-optimized 316L stainless steel hip stems, based upon stochastic porous structure and a selectively hollowed approach. Concerning the hollowed stem, stress shielding of the femoral bone was reduced by 15% in Gruen zone 6, and by 25% in Gruen zone 7. Both optimized stems had an approximate 40% reduction in stiffness, when compared to the solid stem.

With the advanced development of metal additive manufacturing, various porous hip stems were developed using lattice structures. Cortis et al. [17] used a body–centered–cube (BCC) unit cell to design and manufacture a Ti6Al4V porous hip stem, successfully reducing stress shielding up to 11% and 25% in Gruen zone 6 and 7, respectively. Similarly, Mehboob et al. [18] developed a porous hip stem with body-centered-cube (BCC) unit cell having a shell thickness of 0.5–2 mm. They showed that the stress shielding can be reduced by 28%, having a Soderberg FoS of 1.1. Kladovasilakis et al. [19] investigated three differing sheets of triply periodic minimal surface bioinspired scaffolds, namely, Voronoi, Gyroid, and Schwarz Diamond. Topology optimization was performed to develop a Ti6Al4V functionally gradient porous hip stem with the abovementioned structures which demonstrated that the Schwarz Diamond scaffold had the best mechanical behavior with a factor of safety (FoS) of 2.08. Krishna et al. [20] proposed a porous titanium implant, with reduced stiffness (2–45 GPa) via laser engineered net shaping. Naghavi et al. [21,22] performed a detailed mechanical (compression, tension, bending and torsion tensing) and morphological investigation on gyroid and diamond lattice structures suitable for development of porous hip stems to reduce stress shielding.

In addition to the usage of geometric design and lattice structures, the stress shielding effect was also approached by utilizing biomimetic materials. Bougherara et al. [23] designed a biomimetic composite hip prosthesis, based on polymeric composite and a hydroxyapatite-based coating, to obtain properties similar to those of the cortical bone. Their results showed that the femoral bone implanted with a composite structure sustains up to 192% of load than the one implanted with a conventional Ti stem. Oshkour et al. [24] proposed a functionally graded hip implant which consists of stainless steel, titanium alloy, and hydroxyapatite. They subsequently reported a 22% increase in the strain energy density in the proximal femur, resulting in a reduced stress shielding effect. Tavakkoli Avval et al. [25] developed a carbon fiber polyamide 12 hip stem and showed that the bone loss was reduced to 9% when compared to cobalt–chrome (27%) and titanium (21%) hip stem. Anguiano–Sanchez, J. et al. [26] presented a metallic implant with a polymer coating (polyether ether ketone) with the effective von Mises stress increases around 81 to 92% in the cancellous bone.

Generally, previous efforts aimed at reducing stress shielding are mainly derived from modifications in stiffness of the entire femoral stem or stem/bone interface. Although these works are well-designed and the results are meticulously evaluated, the effectiveness of their designs in terms of mitigating stress shielding and extending life span can be hardly verified since none of these implants have been assessed in long-term clinical practice; thus, there is no standard approach to tackle stress shielding effect in the current stage. As concluded in Table 1, only around 20% to 30% of reduction in stress shielding effect with around 40% to 75% of bone resorption reduction compared to generic implants have been achieved in previous research. However, this low value persists, limiting even greater long-term survivorship of hip implants.

Polyether–ether–ketone, commonly referred to as PEEK, is a high–performance thermoplastic. With the advantages of high in-vivo stability and biocompatibility, radiolucency, and favorable mechanical properties, PEEK has been used for trauma and orthopedic, dental, and spinal implants for the last decade and many of them has been commercialized [29,30,31]. Moreover, PEEK biomaterials can exhibit an elastic modulus ranging between 3 and 4 GPa, the modulus can be tailored to closely match femur cortical bone (1–30 GPa) through various processes [32] or even titanium alloy (110 GPa) by preparing carbon-fiber-reinforced (CFR) composites with certain fiber length and orientation [33]. This unique feature makes PEEK a promising low–stiffness biomaterial for orthopedic implants which in turn reduces stress shielding. Several attempts have incorporated PEEK material as part of their implant design. As mentioned, Anguiano–Sanchez, J. et al. [26] developed a PEEK coating with various thicknesses applied in metallic hip implants, the effectiveness in reducing stress shielding is evaluated by finite element analysis. The Epoch hip stem created by engineers at Zimmer, Inc. (Warsaw, IN, Poland) incorporated various polymer coatings including PEEK, PEAK, and PEKEKK on CoCr alloy inner core by extrusion; these designs have been tested through animal studies and human clinical trials with positive outcome [34]. However, most current studies only use PEEK as coating or thin layer, the applications of PEEK in entire hip stem design are still limited; only a few studies exist, such as Oladapo, B.I. et al. [35], who designed porous implants using PEEK and its composites to improve the compatibility of implants, while the stress shielding effect is not examined in their research. Hence, as a novel material choice for entire hip prosthesis design, the stress shielding effect and mechanical behavior of the PEEK hip prosthesis should be deeply investigated. Despite the high price of PEEK which may limit its usage, the excellent properties and potential to increase the service life of the hip stem by reducing the shielding still makes PEEK attractive for orthopedic applications. 

This study proposes a surface porous hip implant using PEEK biomaterial for the entire femoral stem to reduce stress shielding effect. Similar to other uncemented stems, the surface porous structure incorporated in this design is to achieve osteointegration for biological fixation. The ability of pure PEEK implants in reducing stress shielding is examined through experimental tests and FE simulation by comparison with a generic Ti6Al4V implant manufactured with the same geometry mounted in the Sawbone and an artificial intact femoral Sawbone. This study hypothesizes that comparing a generic hip implant to the proposed hip implant will effectively reduce the stress shielding effect and bone resorption. In addition, to evaluate the load-carrying capacity of the PEEK implant, the stiffness of the implant is measured, and the yield and fatigue factor of safety is analyzed.

## 2. Materials and Methods

### 2.1. Hip Implant Design and Manufacture

Figure 1 shows the flow chart of the methodology used to develop a PEEK hip stem that reduces bone resorption as a result of reducing the stress shielding. A customized hip stem was designed by SolidWorks CAD program (SolidWorks Corp., Dassault Systèmes, Waltham, MA, USA) based on the geometry of a large, left, fourth-generation artificial composite femoral bone (Model 3406, Sawbones, Pacific Research Laboratories Inc., Vashon, WA, USA). The hip stem was made to fit as firmly as possible against the cortical bone that surrounds it. The hip stem’s design and dimensions are shown in Figure 2a. The control group for this investigation consisted of a solid Ti6Al4V hip stem that was processed by CNC from a titanium block (BS 2TA11 (Grade 5) Ti6Al4V 20,100,150 mm). Figure 2b illustrates the three distinct regions that make up the PEEK hip stem, which include the solid stem, and anterior and posterior porous surface with 2 mm depth. The same orthogonal rectilinear structure, with unit cell sizes of 0.8 mm, pore sizes of 0.4 mm, and strut sizes of 0.4 mm, was used to construct both the anterior and posterior lattice structures, ensuring the porous part of the scaffolds to have a nominal porosity of 50%. The porous structure is designed to provide biological fixation of the implant through bone ingrowth. The diameter of a sphere that can fit through the biggest pore of a porous material is known as the interconnected pore size. To ensure sufficient bone ingrowth, pore size should be at least 0.1 mm [36]. In addition, higher porosity generally results in greater bone ingrowth properties. To achieve osseointegration, a minimum of 50% porosity is required [37]. In this study, a commercial PEEK raw material (450 G, Lancashire, UK) with a weight-average molecular weight of around 37,000 was used [38]. PEEK filament with a diameter of 1.75 mm was made by a twin-screw extruder. A PEEK hip stem was produced utilizing fused filament fabrication (FFF), also known as fused deposition modelling (FDM) [39], by a commercial 3D printer (Engineer 200, Jugao AM, Xi’an, China) (Figure 2c). Table 2 provides an overview of the 3D printing processing parameters. Compression and tensile tests were previously undertaken to investigate the mechanical characteristics of the PEEK material [40]. The Young’s modulus and yield strength are reported to be 1.69 GPa and 85.5 MPa, respectively.

### 2.2. Experimental Testing

#### 2.2.1. General Strategy 

As shown in Figure 3, an intact femur (Figure 3a), a femur implanted with a Ti6Al4V hip stem (Figure 3b), and a femur implanted with a PEEK hip stem (Figure 3c) are the three configurations that are adopted in this study. The PEEK hip stem is expected to have a lower stiffness than the Ti6Al4V hip stem and promote osseointegration by reason of the porous surface. Hence, the nearby cortical bone would, therefore, experience more natural force distribution mimicking the pre–surgical situation, which theoretically reduces the stress shielding effect and bone resorption. To ensure uniformity in the data collection, a single femur model was used for all configurations. The femur model used in this study will be further explained in Section 2.2.2.

#### 2.2.2. Model Preparation

A left large fourth-generation artificial composite femoral bone—Sawbone (model 3406, Pacific Research Laboratories Inc., Vashon, WA, USA)—was employed in this investigation. The femur bone has an overall length of 485 mm, a canal diameter of 16 mm, a cancellous bone with a density of 0.27 g/mL made of polyurethane foam, and a cortical bone with a density of 1.64 g/mL made of e-glass fibers combined with epoxy resin. These artificial femurs exhibit high repeatability of their geometry and mechanical characteristics, which are validated with human femurs by a range of mechanical tests, according to several publications [41,42,43].

To create a CAD model with the same bone that is utilized in the tests, the artificial femur was scanned using a computed tomography (CT) scanner (Philips Brilliance 64 CT Scanner, AMN, The Netherlands) with a resolution of 0.25–0.30 mm. Every 0.5 mm, a lengthwise CT scan was taken. After that, CT scans were saved in DICOM format and imported into Mimics Medical Imaging Software (The Materialise Group, Leuven, Belgium) to segment the scans and create a 3D model of the normal femur with the cancellous and cortical bone. Following that, the model was exported into SolidWorks CAD (Solid–Works Corp., Dassault Systèmes, Waltham, MA, USA). 

A band saw was used to cut the femur’s distal condyles (by 77 mm) to achieve an overall length of 408 mm. The femur was then vertically potted into a ∅100 × 80 mm steel cylinder, filled with anchoring cement used in industrial construction (Blue Circle Mastercrete Cement, Tarmac Cement & Lime Ltd., Birmingham, UK), resulting in a final working length of 328 mm. After obtaining the surface local strain measurements of the intact bone (described in Section 2.2.3), the femoral head was removed 13 mm above the lesser trochanter at a 45–degree angle (with respect to the horizontal). Almost all the cancellous bone (polyurethane foam) was removed by a surgical femur reamer from the proximal medial and lateral sections of the Sawbone. This was to ensure that the hip stem fitted exactly on the surface of the cortical bone. To guarantee consistent implant location and orientation as well as the right neck offset and length, an X-ray was performed at 62 kV (DigitalDiagnost 2.1.4V22.13.567, Philips Medical Systems DMC GmbH, Hamburg, Germany), as shown in Figure 4. Once the hip stem’s position had been determined to be appropriate, epoxy resin (MC002568, Multicomp, London, UK), which has the same density as cortical bone (1.69 g/mL), was injected into the canal. This resin filled the cavities between the cortical bone and the hip stem, performing a better force distribution from the stem to the surrounding cortical bone, and, thus, enabling a more valid FE model. The resin was then left for 24 h for completed solidification.

#### 2.2.3. Strain Gauge Attachment

To quantify the local surface strain of the artificial femur, ten points of interest (Figure 5a,b) on the femur were fitted with 350 Ω rectangular rosette (45°) strain gauges (FRAB–2–350–23–1LJB–F, Tokyo Measuring Instruments Laboratory Co., Fukuoka, Japan). Strain rosettes were placed in identical locations in each of the three configurations (Figure 3) and were not changed or removed. There were 4 rosettes (L1–L4) on the lateral side and 6 (M1–M4, MX1 and MX2) on the medial side. The lateral rosettes were positioned on the femur at 0 mm (L1), 31.75 mm (L2), 63.5 mm (L3), and 95.25 mm (L4) below the lesser trochanter, whereas the medial rosettes were positioned at 0 mm (M1), 16 mm (MX1), 31.75 mm (M2), 47.75 mm (MX2), 63.5 mm (M3), and 95.25 mm (M4) below the lesser trochanter. All 10 strain rosettes (30 strain gauges) were positioned parallel to the long axis of the femoral shaft. Gauges were wired to form a half bridge completion circuitry, preamplifier, 24-bit analog to digital converters and serial data processor. This system connected with a laptop computer for data storage, and LabVIEW software (2013, National Instruments, Austin, TX, USA) was used for data capture. Strains were recorded as digital counts, with a sensitivity of 545.4 counts/microstrain. To calculate the count difference, the average peak count and zero load count were measured and subtracted. The count difference was divided by the conversion value of 545.4 to obtain the local microstrain value (𝜀𝐴, 𝜀𝐵 and 𝜀𝐶).

#### 2.2.4. Loading and Measurements

Based on ISO 7206:2010, the vertically potted femur Sawbone was distally fixed in all three configurations (Figure 3) at 10° adduction in the coronal plane and 9° flexion in the sagittal plane in an inclined steel platform. The Sawbone femur was mechanically tested in axial compression with increasing load from 500 to 1200 N with a displacement rate of 0.01 mm/s (one–legged stance phase of walking) [44] in steps of 100 N and step time of 25 s. The Sawbone femur was loaded within the linear elastic zone by these applied forces, which is relatively lower than physiological loadings; thus, it may not accurately reflect the physiological loadings associated with varied daily activities. However, it prevented the Sawbone from fracturing during the repeated tests. Ebrahimi [44] has reported that these femurs could break at typical axial stresses as low as 2000 N to 3000 N. To eliminate any potential signal errors and outliers in the data, five repeated measurements were taken for each configuration. These compression tests were conducted on a Zwick machine (Zwick GmbH, Ulm, Germany) with a 5 kN load cell. The order of the tests for the three configurations was 1. intact bone, 2. femur with the Ti6Al4V stem, 3. femur with the PEEK stem. To reduce the potential effect of strength memory in the femur, sufficient interval between the tests for each configuration was added for relaxation of residual stress in the femur.

A limitation of this experiment is that only the local surface strain of 10 selected points in the cortical bone is measured. However, bone resorption secondary to stress shielding is a volumetric phenomenon. To quantify the stress shielding effect and bone resorption, the local strain in the whole volume of identified Gruen zones (Figure 5c) are evaluated via the finite element method (FEM). 

### 2.3. FE Simulation

#### 2.3.1. Assembly of Components

As illustrated in 2.2.2, the femur FE model was created by CT scan imaging and its geometry was modelled in SolidWorks software (Solid–Works Corp., Dassault Systèmes, Waltham, MA, USA). The hip stem, ball, loader, resin, and inclined platform (10° adduction in the coronal plane and 9° flexion in the sagittal plane) were also modelled using SolidWorks. The models for these parts were created based on experimental Vernier calliper measurements of their geometry and dimensions. After assembling in SolidWorks, all the models were exported in Parasolid file format (.x_t) and processed in Abaqus software (version 2019, Dassault Systèmes Simulia Corp., Waltham, MA, USA).

#### 2.3.2. Material Properties and Meshing

The manufacturer’s data sheet was used to determine the cortical and cancellous material properties of the artificial femur. Material properties of all parts were assumed to be elastic, linear, and homogenous isotropic. These material properties were assigned to the cortical shell (E = 16.7 GPa, ν = 0.3 [44]), distal cancellous bone (E = 0.155 GPa, v = 0.3 [44]), loader and ball (E = 200 GPa, v = 0.3, stainless steel), proximal resin (E = 2.5 GPa, v = 0.3), bone cement (E = 3 GPa, v = 0.3), Ti6Al4V hip stem (E = 110 GPa, v = 0.3), and PEEK hip stem (E = 1.69 GPa, v = 0.35).

A 10–node quadratic tetrahedron (C3D10) was used to mesh each component of the FE model (ABAQUS). A mesh convergence study was performed for the cortical bone model using twenty–five mesh densities. Elements of roughly consistent size were applied across the model at each degree of mesh density, with specific element sizes ranging from 0.1 mm to 3 mm The solution converged when the difference in the reported results did not exceed 5% when the number of elements were doubles. [45]. It was demonstrated that the results were converged within 5% with an element size of 0.65 mm. In the Gruen zone regions (Figure 5c) where accuracy is vital for experimental validation and future prediction, manual mesh seeding was carried out to increase the number of elements with element size of 0.5 mm. The results for the cortical, distal cancellous, proximal resin, loader, ball, and hip stem converged with about 1.76, 0.41, 0.38, 0.22, 0.1, and 9.1 million elements, respectively. 

#### 2.3.3. Loading and Boundary Conditions

The contact conditions of the ball–hip stem, hip stem–resin, resin–cortical, and distal cancellous–cortical surfaces were identified as tied contact, while for the loader–ball surface, frictional contact with coefficient of friction at 0.2 was applied. The distal portion of the femoral bone was fixed support with an “encastered” method to prevent movement in all directions (Figure 6). For a valid comparison, the same loading and boundary conditions as those used in the experimental compression test were implemented in FEA. The complete femur was loaded to a maximum of 1200 N while being positioned at 10° medial and 9° anterior to the femur axis in accordance with ISO 7206–4:2010. The comparison and validation of the data from the FE model and the experiment are carried out using Bland–Altman plot. After validating the intact femur, the simulating loads were increased to ISO 7206 standard with 2300 N and high physiological loading activity, jogging with 4839 N [46] in each FE model configuration to predict the volume stress and strain in different Gruen zones of the cortical bone. Stress shielding effect and bone resorption in both Ti6Al4V and PEEK hip stems were analyzed from these FEA results with a load of 2300 N.

#### 2.3.4. Stress Shielding and Bone Resorption Measurement

According to Wolff’s Law, bone remodeling takes place to adapt its structure and geometry to the changes in external loads acting on it. Bone density will decrease when stress is reduced whilst rising if loads are exerted at a high level. Stress shielding is evaluated by comparison of Von Mises stress in the cortical bone between intact ($\langle \sigma^{Intact}\rangle$) and implanted ($\langle \sigma^{Implanted}\rangle$) femurs [47]. Fraldi and Esposito [47], defined Stress Shielding Increase (SSI) as the percentage difference of the Von Mises stress of cortical bone in the intact and implanted femur as shown below:(1)$$\mathrm{Stress}\;\mathrm{Shielding}\;\mathrm{Increase}\;\left(\mathrm{SSI}\right)=\frac{\langle \sigma^{Intact}\rangle -\langle \sigma^{Implanted}\rangle }{\langle \sigma^{Intact}\rangle }\;$$
(2)$$\langle \sigma^{Intact}\rangle \;=\frac{1}{\sum_{e}V_{e}}\underset{V_{e}}{\sum }\overset{}{\underset{V_{e}}{\int }}\left(\sigma_{e}^{Intact}\right)dV\;$$
(3)$$\langle \sigma^{Implanted}\rangle \;=\frac{1}{\sum_{e}V_{e}}\underset{V_{e}}{\sum }\overset{}{\underset{V_{e}}{\int }}\left(\sigma_{e}^{Implanted}\right)dV\;$$
where *σ^Intact^* and *σ^Implanted^* are Von Mises stress before and after THA at the centroid of each element in the femur respectively, and *V_e_* is the volume of each element. SSI reflects the changes in local stress in a region after implantation. A positive SSI implies that the local region experiences less stress than pre-surgical conditions, which induces stress shielding; a negative SSI, on the other hand, suggests a rise in local stress or potential stress concentration [48].

Bone loss secondary to stress shielding is evaluated with strain adaptive reconstruction theory. Based on Huiskes’ bone adaptation law, the bone remodeling rate can be described by the following formula [49]:(4)$$\frac{d\rho }{dt}\{=0\begin{array}{c} >0,when\;S>\left(1+s\right)\cdot S_{intact}\;\\ ,\;when\;\left(1-s\right)\cdot S_{intact}\leq S\leq \left(1+s\right)\cdot S_{intact}\\ <0,when\;S<\left(1-s\right)\cdot S_{intact}\; \end{array}\;$$
where $\frac{d\rho }{dt}$ is the bone density change rate, S is the local strain energy (U) per unit of bone mass (*ρ*), and $S_{intact}$ is the value of S before implantation. In this study, only the condition $S<\left(1-s\right)\cdot S_{intact}$ is employed since it reflects the bone resorption. It has been shown that not all the changes in local strain energy will cause bone remodeling, a certain range of overloading or underloading is tolerated. This range is defined as dead zone (*s*), and 0.6 is the typical value for dead zone width obtained by Turner et al. [50] from 2 years of clinical densitometry measurements (DEXA) by measuring the principle compressive strain before ($\varepsilon_{Intact}$) and after ($\varepsilon_{Implanted}$) implantation, the strain energy ratio is calculated as:(5)$$\frac{S_{Implanted}}{S_{Intact}}={\left(\frac{\varepsilon_{Implanted}}{\varepsilon_{Intact}}\right)}^{2}\;$$
where $S_{Implanted}$ is the strain energy after implantation and $S_{Intact}$ is the strain energy before implantation. Based on Equations (4) and (5), the resorbed bone mass fraction $m_{r}$ of local point b can be calculated from:(6)$$m_{r}\left(b\right)=\frac{1}{M}\overset{}{\underset{V}{\int }}f\left({\left(\frac{\varepsilon_{Implanted}{\left(b\right)}^{2}}{\varepsilon_{Intact}{\left(b\right)}^{2}\cdot \left(s-1\right)}\right)}^{2}\right)\rho \left(b\right)dV\;$$
where *M*, *ρ* and *V* are the mass, density, and volume of intact bone, respectively. f(x) is a resorptive function equal to unity when *x* < 1, while equal to 0 when *x* ≥ 1. x < 1 means the stress shielding effect at point b is large enough to induce local bone resorption, and *x* ≥ 1 suggests no bone remodeling taking place at this point. All measurements of Von Mises stress and principle compressive strains were obtained and compared from each predefined Gruen zones (Figure 5c) which are commonly used clinically to assess the performance of THA.

### 2.4. Mechanical Testing of the Protheses

#### 2.4.1. Static Tests

In static tests, the Ti6Al4V and PEEK stems were fixed in accordance with ISO 7206–4:2010 using a PMMA bone cement (Simplex P, Stryker Corp., Mahwah, NJ, USA) inside of a spherical mild steel container that was 50 mm in diameter and 58 mm in height. To avoid cracking of bone cement during the test and provide more support to the samples, an epoxy resin (MC002568, Multicomp, London, UK) was applied on the top of the bone cement to cover all the interfaces. Ti6Al4V and PEEK stems were loaded up to 1200 N with loading rate at 0.01 mm/s by a Zwick machine (Zwick GmbH, Ulm, Germany). The stiffness of these two stems were estimated from the slop of the load–displacement graph within the visible linear region. 

#### 2.4.2. Yield and Fatigue Factor of Safety

To evaluate the load-bearing capacity of PEEK stem in long-term performance, the factor of safety (FoS) evaluation is performed. FoS expresses how much stronger a system is than it needs to be for an intended load. Generally, FoS is required to be at least larger than one to ensure the security of the loading system. In the static loading system, data from FEA, with 2300 N (ISO 7206–4) and 4883 N (force while jogging) loading conditions, were utilized to calculate the yield factor of safety ($FoS_{yield}$) as shown in Equation (7).
(7)$$FoS_{yield}=\frac{\mathrm{Yield}\;\mathrm{stress}\;}{\mathrm{Maximum}\;\mathrm{stress}}$$
where the yield strength of PEEK material is 85.5 ± 0.63 MPa. These data were acquired from standard tensile tests, in which ASTM D638–14 was followed. 

In service conditions, the implanted stem should undergo cycling loads due to the daily activities of the patient. Hence, the study on the fatigue life of the PEEK stem is of significance. To numerically analyze the fatigue factor of safety of the proposed stem, Soderberg fatigue theory, which is considered to be an accurate and conservative approach for medical research [51], is used. Stress ratio (R) = 0.1, with the minimum stress $\left(\sigma_{min}\right)$ collected at 230 N and maximum stress $\left(\sigma_{max}\right)$ collected at 2300 N in one loading cycle, is applied in this case. The mean stress $\left(\sigma_{m}\right)$ and alternating stress $\left(\sigma_{a}\right)$ are measured using Equations (8) and (9) respectively.
(8)$$\sigma_{m}=\frac{\left(\sigma_{max}+\sigma_{min}\right)}{2}$$
(9)$$\sigma_{a}=\frac{\left(\sigma_{max}-\sigma_{min}\right)}{2}\;$$

Based on the calculated mean stress and alternating stress of each point, Soderberg equations Equation (10) were used to calculate the fatigue FoS for the whole regions of the PEEK implant.
(10)$$FoS_{Soderberg}=\frac{1}{\frac{\sigma_{a}}{\sigma_{N}}+\frac{\sigma_{m}}{\sigma_{y}}}\;$$
where $\sigma_{y}$ and $\sigma_{N}$ are the yield strength and the endurance limit of the PEEK, respectively. ISO 7206–4:2010 requires the hip stem to withstand $5\times 10^{6}$ cycles before fatigue. Hence, the endurance limit of the PEEK at $5\times 10^{6}$ cycles to failure should be used. Based on this requirement, the endurance limit of the PEEK at 70 MPa is selected for this study. This data is obtained from literature reported by Pastukhov et al. [52]. For data analysis, the points in the PEEK stem whose $FoS_{Soderberg}$ is larger than one, are considered to have an infinite lifespan in our simulation. However, if any point has $FoS_{Soderberg}$ that less than one, the whole stem is likely to fail due to fatigue.

## 3. Results and Discussion

### 3.1. Experimental Testing and Validation of the FEA Model

Both experimental testing and FEA are conducted to assess the surface von Mises stress and compressive strain for the three configurations, including the intact femur, the femur implanted with a Ti6Al4V hip stem, and the femur implanted with a PEEK hip stem. The local strain gauge results for experimental and FEA studies are shown in Figure 7a and 7b, respectively. The von Mises stress ratio is defined as the von Mises stress of implanted femur divided by that of intact bone on the same point, which suggests the change in stress in each point of interest after implantation with Ti6Al4V and PEEK stems. From the experimental data, it can be seen that for most sites, the femur with the PEEK stem has a higher stress ratio compared to the femur with the Ti6Al4V stem, which indicates that the femur with the PEEK stem would undergo more load compared to its counterpart in loading condition, and, thus, mitigates the stress shielding effect. As illustrated in Figure 7a,b, strain gauges MX2, M3, M4, L3, and L4 (highlighted with a red dotted box) show that the Ti6Al4V stem has a higher stress ratio than 100% when compared to the intact bone. This suggests that there is no stress shielding effect on these points. These findings align with the results in the literature, which report limited stress shielding in distal regions [53]. For the FEA study, similarly, there is no significant change in stress in MX2, M3, M4, L3, and L4 after implantation of both stems. For other points (M1, MX1, L1, and L2), the stress ratio for the PEEK stem is much higher (by approximately 40%) than that for the Ti6Al4V stem. 

When comparing the experimental and FEA stress ratio of the PEEK stem, it has shown that the experimental stress ratio of the PEEK stem in strain gauges M1 and L1 are significantly larger than the FEA stress ratio. On the other hand, the stress ratio of PEEK stem in strain gauges M2, L2, and L3 have extremely low stress ratio in the experimental study in comparison to the FEA stress ratio. These deviations between the experimental and FEA study are caused due to slight geometrical mismatch of the printed PEEK stem with the Sawbone. Limitation in manufacturing precision is known to be the cause of this geometrical mismatch which prevents full contact of the PEEK stem to the surrounding cortical bone. Figure 4 shows the X-ray images of implanted Ti6Al4V and PEEK stems inside the Sawbone. Although both Ti6Al4V and PEEK stems were produced based on the same CAD model, the Ti6Al4V stem is tightly in touch with the Sawbone. While in the PEEK stem, a small gap is visible between the stem and the Sawbone cortical shell in the distal part, in M2, L2, and L3 regions. The loose contact in M2, L2, and L3 leads to the low surface stress in these points and large stress concentration in M1 and L1, which carry the load that is supposed to be taken from the loosely contacted part of the femur.

Except for the points which were affected by the gap in the PEEK stem model, experimental and FEA studies exhibited a similar pattern. FEA validation was undertaken by comparing the experimental and FEA results via the Bland–Altman plot. As shown in Figure 8, the dots present the mean and difference (bias) between the von Mises stress obtained by experiment and FEA from each point of interest. Since all the dots are within the 95% confidence interval, a good agreement between the experimental test and FEA study is confirmed; thus, the validation of the FE model of the intact femur is proved.

### 3.2. FEA Results for Simulating Load at 2300 N 

The simulating load in this part is increased to 2300 N which corresponds to ISO 7206:2010 standard. As shown in Figure 9a, the equivalent stress (von Mises stress) and compressive strain (E33) for the three configurations are assessed. For stress evaluation, Ti6Al4V and PEEK stem share similar stress distribution in the neck. The stress concentrates on the lateral and medial side surface of the neck. The equivalent stress on the medial and lateral neck is up to 112 MPa and 86 MPa for the Ti6Al4V implant, and 82 MPa and 74 MPa for the PEEK implant. However, the stress distribution in the distal parts of two implants are different. The distal part of the Ti6Al4V implant attracts more stress, which goes up to 73 MPa, while the PEEK stem shows little stress as low as 2 MPa in the same region. As for the stress distribution in the cortical shell, different from the intact bone model, the femoral shafts of both PEEK and Ti6Al4V configurations take much more stress than the proximal femur, which suggests that the proximal femur is the potential location of the stress shielding. This finding is consistent with previous research [54]. In addition, it is noticeable that the stress distribution in the intact femur is more similar to that of the PEEK stem than that of the Ti6Al4V stem, especially in Gruen zone 1, 2, 6, and 7. It implies that the femur with PEEK stem has a more natural stress distribution compared to that with Ti6Al4V stem, which may contribute to the reduction in the stress shielding effect.

For strain analysis (Figure 9b), positive strain indicates the extension in the local point which is motivated by tensile stress; conversely, a negative value indicates the compression motivated by compressive stress. There is little micro-strain observed in the Ti6Al4V stem due to the high stiffness of Ti6Al4V material, while the PEEK implant with relatively low stiffness shows a large negative strain on both sides of the neck with a large positive strain in the middle. For the strain in the femur, the femur head and lateral neck in intact bone show large compressive and tensile strain respectively. Similar to stress distribution, all the configurations have large strain distributed in the distal femur, with tensile strain on the lateral side and compressive strain on the medial side. Unlike intact bone and PEEK stem, there is a limited strain in Gruen zone 1, 2, 6, and 7 for Ti6Al4V stem, which is consistent with the finding in stress evaluation.

### 3.3. Stress Shielding and Bone Resorption Evaluation 

The stress shielding effect is evaluated through the averaged stress shielding increase (SSI) in each Gruen zone. The comparison of stress shielding for the Ti6Al4V and PEEK stems is discussed in this section. The von Mises stress obtained from FEA with 2300 N load is sorted and averaged based on each Gruen zone (Figure 10a). Then, the SSI is calculated with the results presented in Figure 10b. Large SSI indicates the large reduction in equivalent stress in the femur after THA, which may result in bone loss with this underload pattern. For all the Gruen zones, the SSI value for the PEEK stem is much lower (by approximately 40%) than that of the Ti6Al4V stem, except for Gruen zone 4 where the two stems share a similar value with no stress shielding effect. 

As shown in Figure 10a, the von Mises stress ratio of the PEEK stem has less than 21% of the difference from that of intact in all Gruen zones, which indicates that a femur with the PEEK stem has a relatively natural stress distribution. There is no stress shielding effect on the PEEK stem in Gruen zone 1 to 4 since the SSI value in these zones is below 0%. The SSI value in Gruen zone 4 for both Ti6Al4V and PEEK stems is negative, which suggests that no stress shielding showed in this region even with generic Ti6Al4V stem. This finding corresponds to previous research on a porous Ti femoral stem [27]. 

For better contrast, the total SSI for both implants is calculated according to their SSI value and volume fraction of each Gruen zone. The Ti6Al4V and PEEK stems have a total SSI value of 31.60% and −1.15%, respectively. Based on the total SSI value, it is concluded that the PEEK stem has a 104% lower (SSI) value when compared to the Ti6Al4V stem.

To quantitatively assess the bone loss for these two stems and predict the location of bone loss, the resorbed bone mass fraction $m_{r}$ is calculated for both Ti6Al4V and PEEK stems in each Gruen zone (Figure 10c). For the Ti6Al4V stem, Gruen zone 7 is expected to have the largest bone resorption with about 89% mass reduction, Gruen zone 1 and 6 share similar bone resorption at approximately 73%. Gruen 2, 3, and 5 have relatively less bone loss with a mass reduction between 10% to 20%. For the PEEK stem, only Gruen 1 and 7 show significant bone resorption at 35% and 26%, respectively, while the rest of the regions have a negligible bone loss at below 3%. This trend corresponds to the stress shielding evaluation, which suggests the stress shielding effect only take place in Gruen zone 1 and 7 for the PEEK stem. There is a limited mass reduction in Gruen zone 4 for both stems, which justifies the previous conclusion that no stress shielding effect is observed in Gruen zone 4. Generally, the PEEK stem is expected to induce much less bone resorption than the Ti6Al4V stem in all regions except Gruen zone 4, with the total bone loss at 11.6% versus 41.4% for Ti6Al4V stem. Furthermore, according to the position of each Gruen zone, the proximal femur, including Gruen zone 1 and 7, is more vulnerable to bone loss compared to the lower part of the femur, including Gruen zone 3, 4, and 5. 

To further evaluate the effectiveness of the PEEK stem in mitigation of bone resorption, the bone loss reduction for all pre-defined Gruen zones with PEEK stem are calculated and presented in Figure 10d. Apart from Gruen zone 4, which does not show any stress shielding, the PEEK stem is expected to reduce the bone loss in the Ti6Al4V stem by at least 50%. It is remarkable that the bone loss reduction with PEEK stem almost reaches 100% for Gruen zone 5 and 6, which suggests that the bone loss in the medial side of the femur with Ti6Al4V stem can be effectively minimized by using PEEK stem. By combining the results for all the regions together, a 72% reduction in bone loss secondary to stress shielding for the PEEK stem is obtained. To briefly conclude, the proposed implant displayed confidence in reducing the stress shielding effect and bone resorption compared with generic hip implant, thus, the hypothesis is accepted. With the proposed implant, the stress shielding effect can be completely eliminated in most regions with an average SSI even below 0. Compared to the generic implant, the stress shielding reduction is around 104%, which is far beyond the value of 20 to 30% reported in previous studies. The total bone loss reduction is around 72%, which is at a high standard compared to the 40% to 75% reported in past studies.

### 3.4. Stem Stiffness

To ensure a sufficient and safe lifespan and reduce the rate of revision surgeries, the mechanical properties of the PEEK stem should be carefully evaluated. In this study, compression tests are conducted for both PEEK and Ti6Al4V stems to examine the stiffness of the stems. Figure 11 shows the load–displacement chart and stiffness line of the Ti6Al4V stem, PEEK stem and human intact femur range [55]. The Ti6Al4V stem has a higher stiffness at 2.758 kN/mm as the intact femur at 1.446 and 1.163 kN/mm for males and females, respectively. Reversely, a relatively low stiffness at 0.276 kN/mm is obtained for the PEEK stem, which is below the range of stiffness for the intact femur. 

In a component that contains materials with different stiffness, the stiffer material is expected to attract higher stress than a less stiff material when subject to load. Based on the stiffness results, the femur with the Ti6Al4V stem is expected to have lower stress after THA, since the Ti6Al4V stem is stiffer than the femur; the femur with PEEK stem should have the opposite case, in which higher stress after THA is expected. This finding corresponds to the results in the evaluation of stress shielding increase, in which the total SSI for Ti6Al4V stem is positive and the one for PEEK stem is negative. 

### 3.5. Yield and Fatigue Factor of Safety Evaluation

Two configurations are incorporated in this study including: 1. PEEK stem fixed in the sawbone, which mimics the post-surgical loading conditions, and 2. PEEK stem fixed in the cement, which complies with with ISO 7206–4:2010. In static analysis, the yield factor of safety for each element in the PEEK stem from the FE model is studied. The distribution of von Mises stress and yield factor of safety for the first and second configurations with an axial load of 2300 N is presented in Figure 12a and Figure 12b, respectively. For the first configuration, the neck of the PEEK implant suffers from large equivalent stress, especially on the surface of the medial and the lateral side of the neck. The highest stress is shown in the medial side of the neck at 82.3 MPa, which is just below the yield strength of PEEK material at 85.5 MPa. By transferring the scale of the measurement into yield FoS, we can see the FoS for the distal part of the implant is larger than 20, which suggests that this part of the PEEK implant is safe under a force of 2300 N. However, the neck part of the implant shows an FoS as low as 1.04, which is a little more than 1. For the second configuration, the stress distributed on the neck is slightly lower than the first configuration with stress at 67.4 MPa. The highest stress concentration is shown on the distal part of the stem with stress at 76 MPa and FoS at 1.13, which is close to one. These results indicate that, in our simulations, the proposed stem is likely to survive when the force applied is below 2300 N, whereas any axial force larger than 2300 N may cause the implant failure by yield. 

Nevertheless, when the axial force increases to 4883 N, the lowest yield FoS in the PEEK implant of the first and the second configurations is decreased to only 0.441 and 0.483, respectively. This means that the PEEK implant is not able to withstand the load at 4883 N. Given that 2300 N and 4883 N are the force applied in the femur while walking and jogging respectively, the proposed implant is capable of supporting walking and other less intense activities, but is far from satisfactory when considering jogging. 

In dynamic analysis, the fatigue property of the PEEK implant is evaluated through Soderberg theory. According to the Soderberg equation, a point in the hip stem will not undergo fatigue failure if the alternating stress and mean stress of this point both sit below the Soderberg line. As shown in Figure 12c, all the datapoint of elements in the second configuration sit in the safe range ($FoS_{Soderberg}$ larger than 1), with the lowest $FoS_{Soderberg}$ at 1.023. For the first configuration, most of the data points are located below the Soderberg line, while there are small portions of the data points, which are not within the safe range. From the FEA data, there are around 0.05% of the elements in the PEEK stem have the $FoS_{Soderberg}$ less than 1. It indicates that there is 0.05% volume of the PEEK implant is likely to fail by fatigue after a long time of use under the post-surgical loading conditions. These elements are distributed in both medial and lateral sides of the implant’s neck, which suggests the potential location of crack initiation.

### 3.6. Limitations

The current study is limited by using isotropic mechanical properties for the bone model. Future study should take a deeper look at the actual anisotropic mechanical properties of bone for a more accurate modelling. The bone resorption evaluation in this study is an estimation of bone adaption when bone remodeling is reaching equilibrium at its maximum value, which may not reflect the bone resorption immediately post the operation and may not be comparable to the clinical data. At present, there are no widely accepted in vitro biomechanical models available that can represent bone resorption process. To validate the bone resorption phenomenon of the proposed implant, long term in vivo or animal models are needed.

## 4. Conclusions

This study investigated the stress shielding effect and bone resorption of additively manufactured PEEK hip prostheses compared with generic Ti6Al4V hip prostheses through resistance strain gauge and FE simulation in ABAQUS. There is no significant difference between the stress data obtained from the strain gauge and the simulation, which confirmed the feasibility of FEA in stress shielding evaluation. The femur with the PEEK implant showed a more natural stress distribution than the femur with the Ti6Al4V implant, which is comparable to a pre-surgical condition. For both implants, stress shielding is reported in the proximal femur, while no apparent stress shielding is found in the lower part of the femur. The proximal femur with Ti6Al4V implant showed up to 68% of stress shielding increase, whereas this value is only around 20% in the case of the PEEK implant. A 104% reduction in stress shielding increase is obtained for the PEEK implant compared with the Ti6Al4V implant. In the long-term use, the PEEK implant and the Ti6Al4V implant are expected to induce 11.6% and 41.4% of bone resorption, respectively. The PEEK implant is able to reduce the bone loss in the femur caused by implantation of the generic implant by 72%. This effect is extremely apparent in the medial region of the femur, in which close to 100% of bone loss reduction is reported.

The mechanical properties of the PEEK implant were studied, which includes the stiffness tests, and evaluation of the factor of safety obtaining from both static and dynamic analysis. The stiffness of the PEEK implant (0.276 kN/mm) is below the range for the intact femur (1.163 to 1.446 kN/mm). This explains the results in stress shielding evaluation, in which the stress distributed in the femur with the PEEK implants is even higher than in pre-surgical conditions. In static analysis, the PEEK implant is proved to be safe at a load equivalent to walking status, but likely to fail with a larger load at the neck in more intensive activities, e.g., jogging. In dynamic analysis, 0.05% volume of the PEEK implant, which is distributed in the near-surface region of both medial and lateral sides of the neck, is expected to fatigue in the required lifespan. Future work can cover tailoring in mechanical properties in terms of yield strength and endurance limit of the PEEK material since the mechanical properties of the proposed implant are slightly below the industrial standard. A possible strengthening method refers to heat-treatment, and incorporation of reinforcing fibers. Another future direction could focus on the experimental testing on fatigue behavior of the proposed implant to further confirm its durability in long-term use.

## Figures and Tables

**Figure 1 polymers-14-04600-f001:**
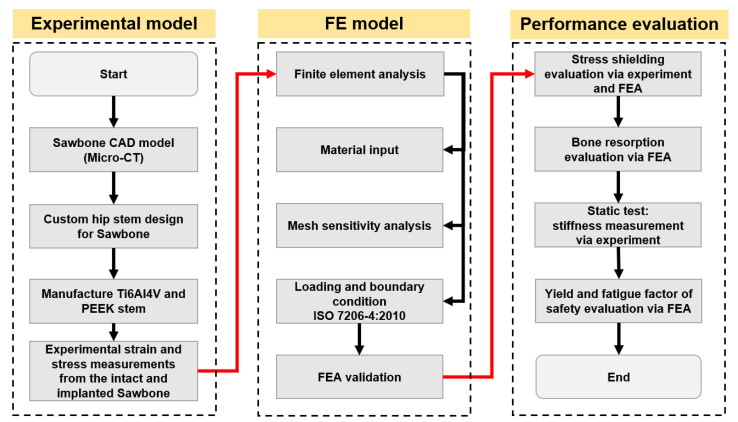
Flow chart of general strategies used in this study.

**Figure 2 polymers-14-04600-f002:**
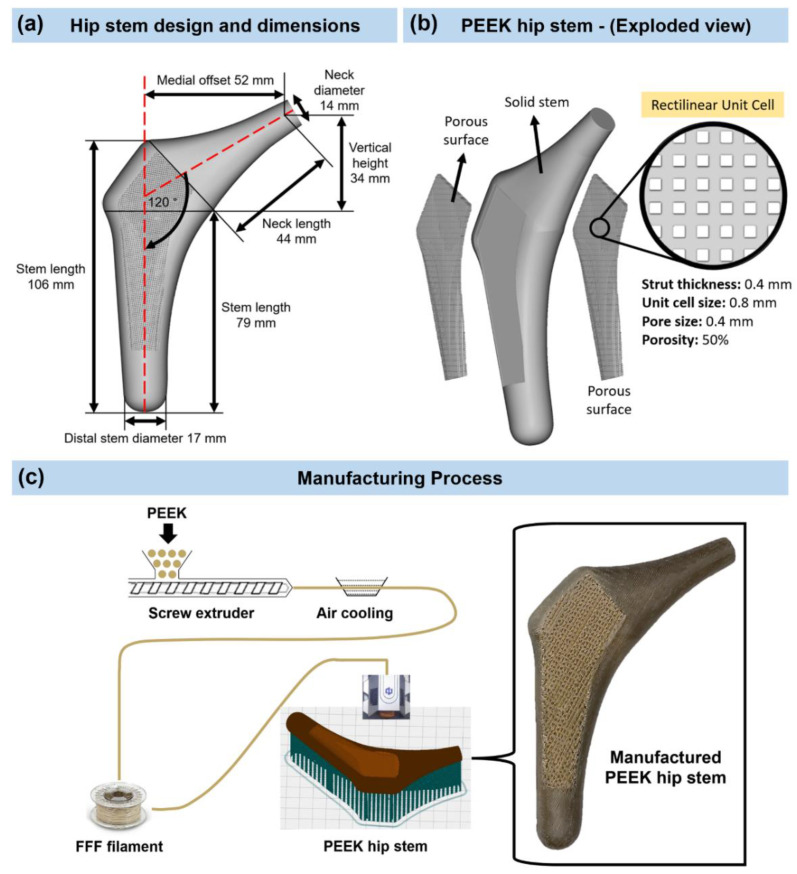
(**a**) Design and dimensions of the hip stem. (**b**) Isometric exploded view of the PEEK hip stem. (**c**) Manufacturing process and 3D printed PEEK hip stem with porous surface rectilinear scaffolds unit cells.

**Figure 3 polymers-14-04600-f003:**
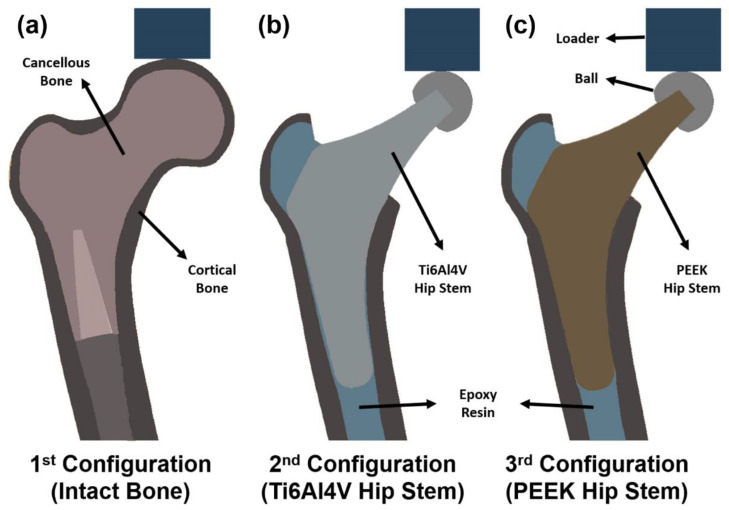
Section view schematic of (**a**) intact femur model as 1st configuration, (**b**) implanted Ti6Al4V hip stem femur model as 2nd configuration, and (**c**) implanted PEEK hip stem femur model as 3rd configuration.

**Figure 4 polymers-14-04600-f004:**
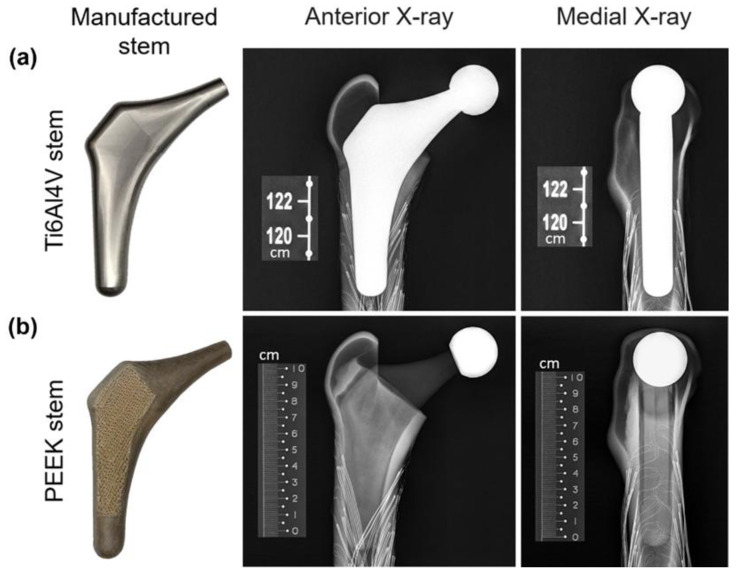
X-ray images of the Ti6Al4V (**a**) and PEEK (**b**) stems implanted in the Sawbone.

**Figure 5 polymers-14-04600-f005:**
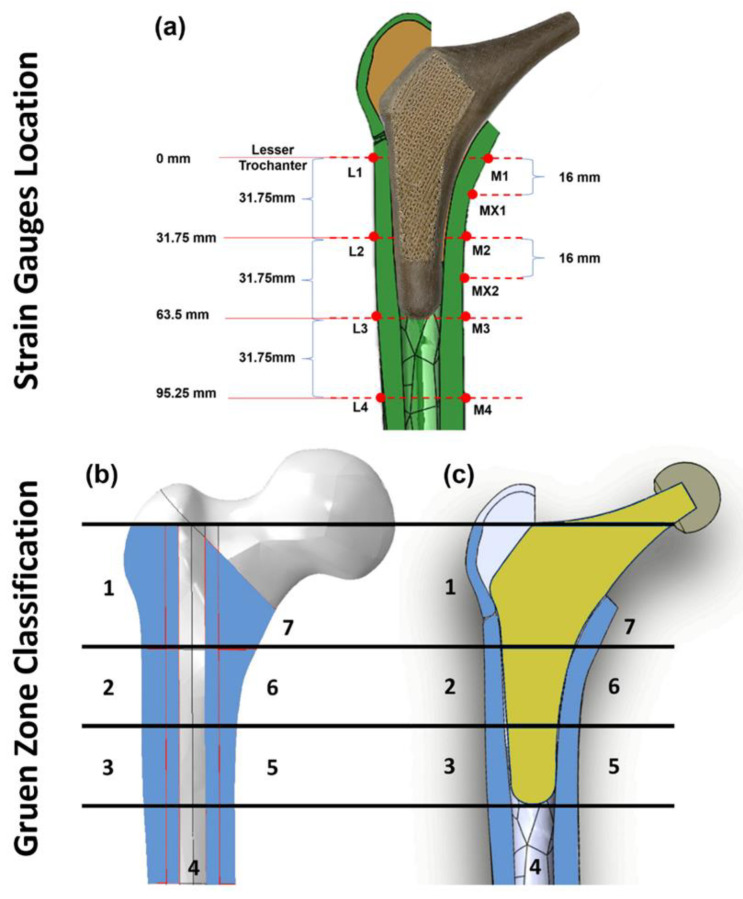
Illustration of strain gauge locations on the surface of the cortical bone (**a**), Gruen zones (1–7) on intact femur bone model (**b**), and section view of the implanted hip stem inside the femur Sawbone (**c**).

**Figure 6 polymers-14-04600-f006:**
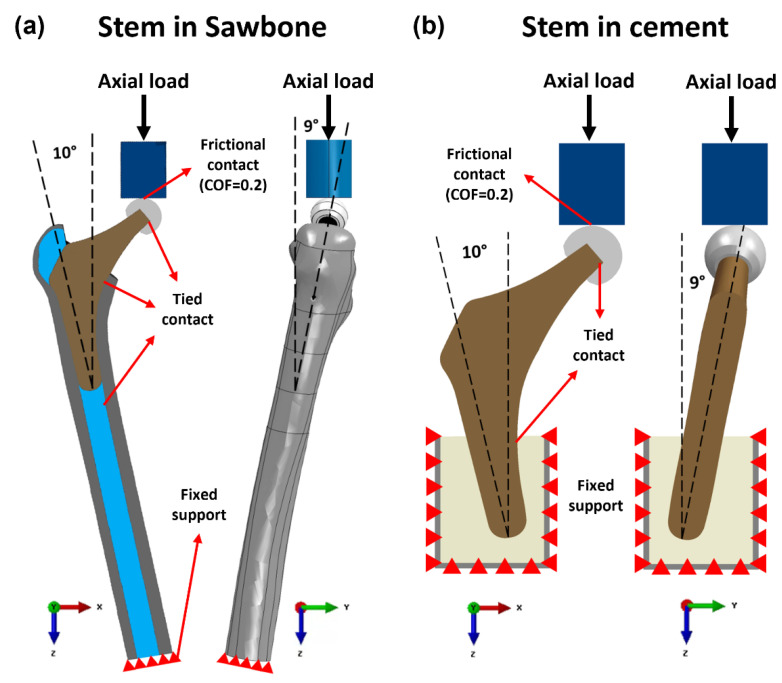
Loading and boundary conditions of the FE model of the Ti6Al4V and PEEK stem fixed in (**a**) Sawbone and (**b**) cement.

**Figure 7 polymers-14-04600-f007:**
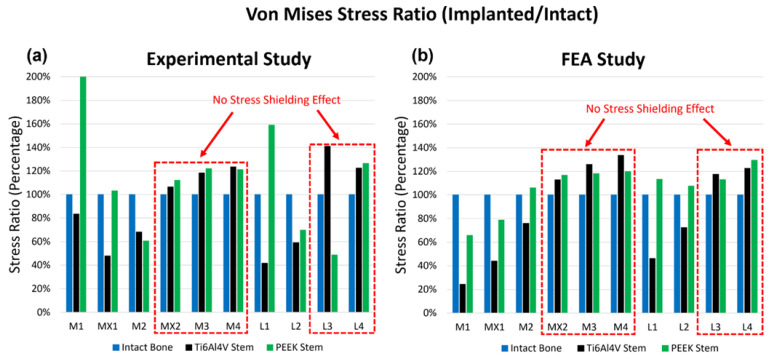
The stress ratio of the 10 points of interest for the intact bone, the Ti6Al4V Stem, and the PEEK Stem, obtained from experimental study (**a**) and FEA study (**b**). The x-axis demonstrates the rosettes strain gauge locations on the cortical bone. There were 4 rosettes (L1–L4) on the lateral side and 6 (M1–M4, MX1 and MX2) on the medial side.

**Figure 8 polymers-14-04600-f008:**
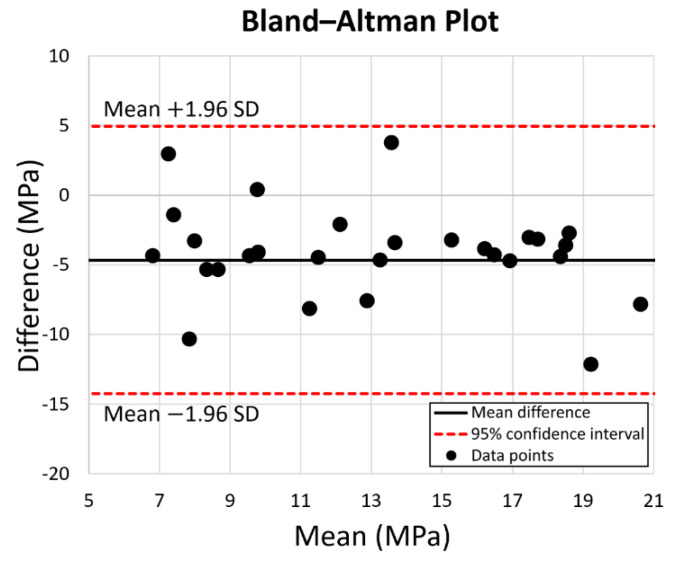
Prediction accuracy by Bland–Altman Plot for the equivalent stress for the 10 points of interest on the intact bone, the PEEK stem, and the Ti6Al4V stem. The red dotted lines represent the 95% confidence interval.

**Figure 9 polymers-14-04600-f009:**
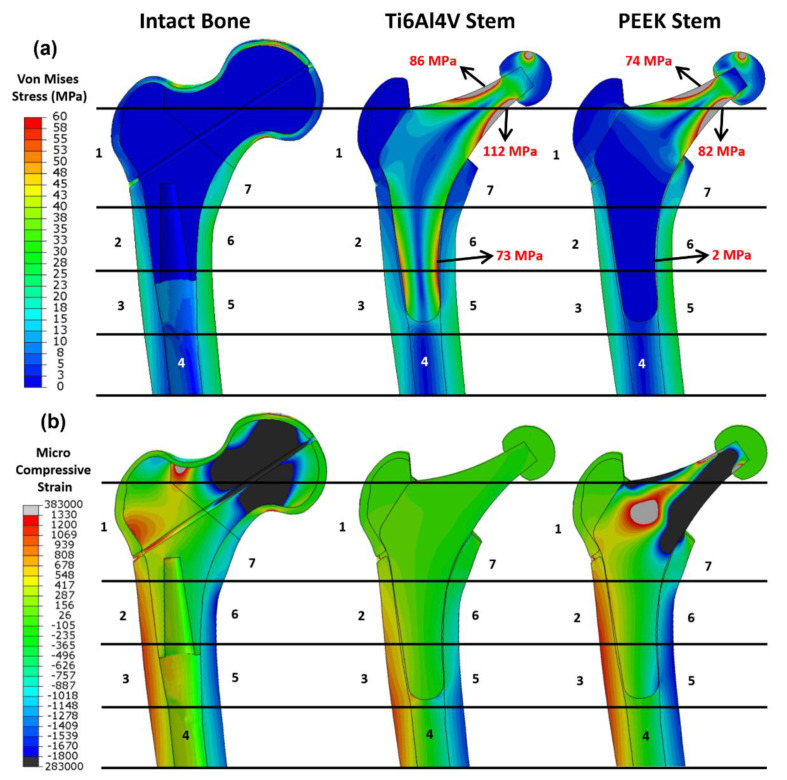
Distribution of the (**a**) von Mises stress and (**b**) the compressive strain for the intact bone, the femur with the Ti6Al4V stem, and the femur with the PEEK stem. The numbers indicate the Gruen zones 1–7.

**Figure 10 polymers-14-04600-f010:**
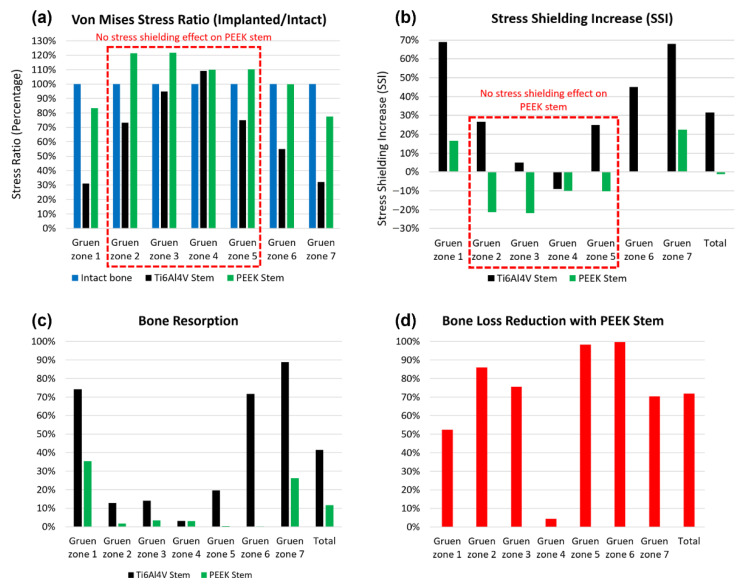
The von Mises stress ratio (**a**), stress shielding increase (**b**), bone resorption (**c**), and bone loss reduction with the PEEK stem compared with the Ti6Al4V stem (**d**) in each Gruen zone from the Gruen zone 1 to 7.

**Figure 11 polymers-14-04600-f011:**
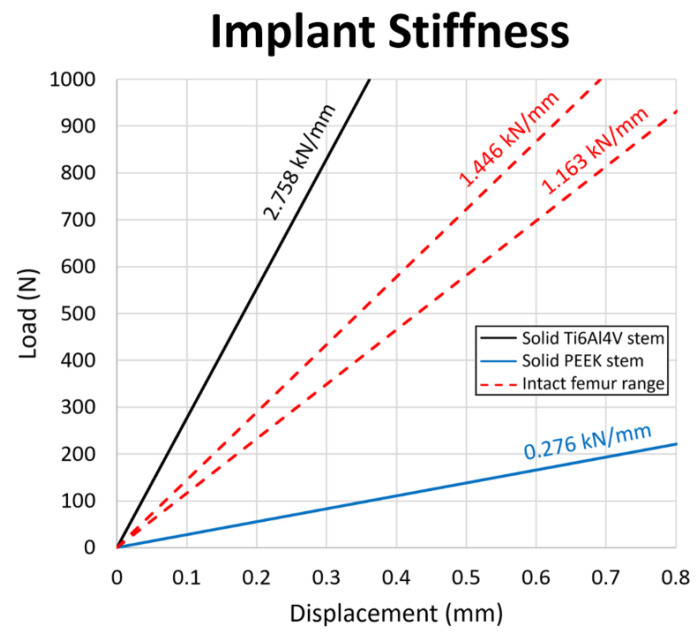
Load–displacement graph for the Ti6Al4V stem, the PEEK stem, and the intact femur. The stiffness value of each configuration is presented above the respective slopes on the diagram.

**Figure 12 polymers-14-04600-f012:**
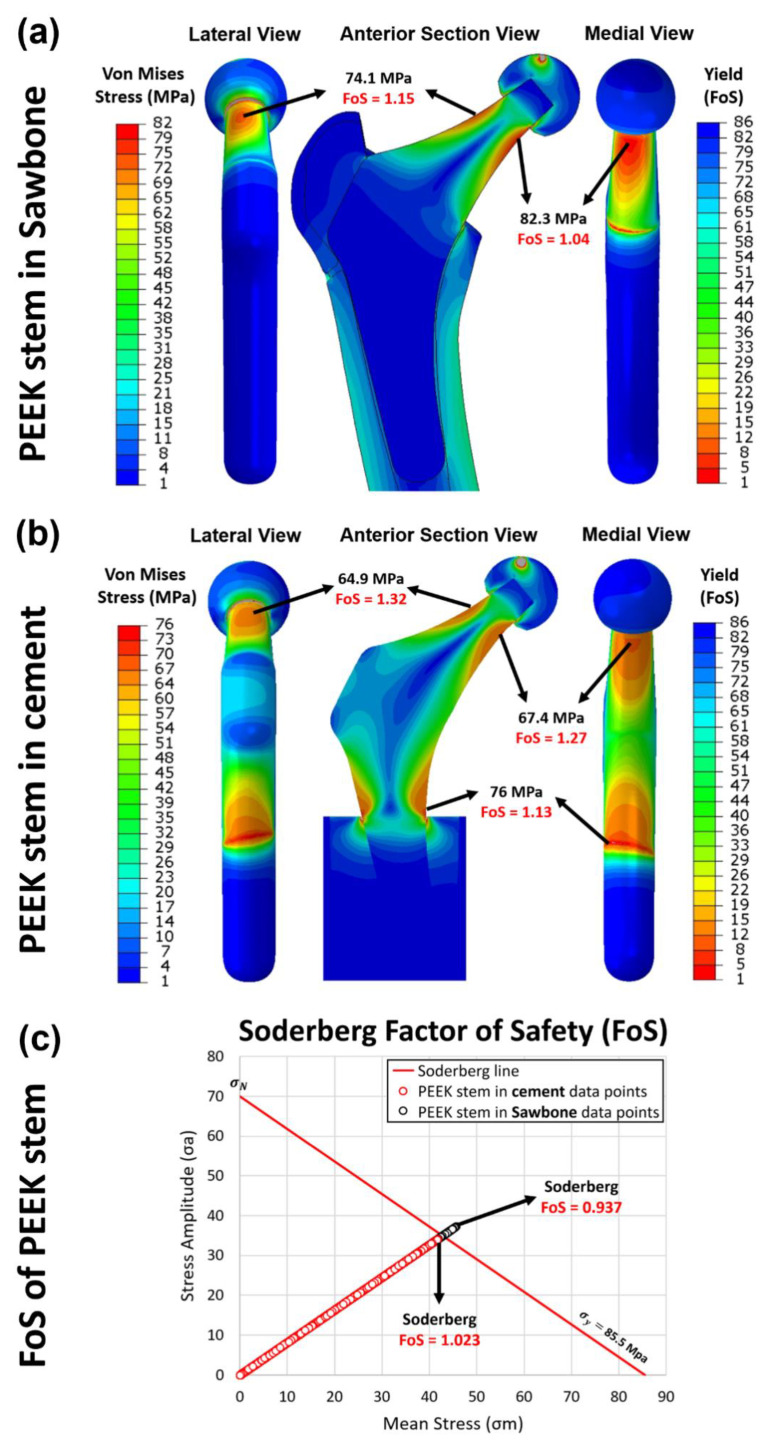
The yield factor of safety distribution in the implanted PEEK stem in (**a**) Sawbone and (**b**) cement, loaded at 2300 N. (**c**) the fatigue factor of safety of each element in the PEEK stem implanted in the Sawbone (black circle) and in the cement (red circle), examined through the Soderberg fatigue theory.

**Table 1 polymers-14-04600-t001:** Stress shielding and bone resorption reduction claimed in previous studies. En dash (–) line indicates an absence of data.

Previous Studies	Stress Shielding Reduction	Bone Resorption Reduction
Sun, C. et al. [14]	32%	40%
Tan et al. [16]	15% and 25% in Gruen zone 6 and 7, respectively.	–
Cortis et al. [17]	11% and 25% in Gruen zone 6 and 7, respectively.	–
Mehboob et al. [18]	28%	–
Arabnejad et al. [27]	–	75%
Wang et al. [28]	–	58.1%

**Table 2 polymers-14-04600-t002:** The process parameters of 3D printing.

Parameters (Unit)	Value
Nozzle Temperature (°C)	420
Ambient Temperature (°C)	20
Nozzle Diameter (mm)	0.4
Printing Speed (mm/s)	20
Layer Thickness (mm)	0.2

## Data Availability

All data are contained within the article.

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
