# Peer review of "Stress Shielding and Bone Resorption of Press-Fit Polyether–Ether–Ketone (PEEK) Hip Prosthesis: A Sawbone Model Study"

_polymers, 2022, doi:10.3390/polym14214600_

Round 1
Reviewer 1 Report
The author uses mechanical tests and FEA to investigate the stress shielding effect and the potential bone resorption of the PEEK implant. The manuscript is complete and informative, and I think it can be considered for publication with the following corrections.
The author mentions the advantages of many PEEK materials. It is true that PEEK has great potential for implant applications, but my concern is, can PEEK materials be widely used in biomedicine or medicine (such as certified by FDA, EMA, NMPA, and so on)? In addition, PEEK may not be good alternative material in terms of price (too expensive?). It is suggested to state it in the introduction or discussion.
Please include the hypothesis of this experiment in the last paragraph of the introduction and state in the discussion whether the hypothesis was accepted or rejected.
The author mentioned that Young's modulus and yield strength of the PEEK materials used are 1.69 GPa and 85.5 MPa, respectively. Generally, Young's modulus of pure PEEK is between 3-6 GPa. Is the PEEK raw material (450G, VIC- 165 TREX, UK) modified or added with other substances to reduce its elastic modulus? In addition, the melting point of PEEK is about 340~380°C, and the nozzle temperature of the FDM 3D printer is 420°C. Will this cause carbonization of the printed PEEK?
The arrange of data, the present of the figures and tables, and the description of the results and discussions are all good. I am honored to review your outstanding work. And it is expected that more research can be proposed to improve the application of PEEK materials. Great Job!
Author Response
Response to Reviewer 1:
Point 1. The author mentions the advantages of many PEEK materials. It is true that PEEK has great potential for implant applications, but my concern is, can PEEK materials be widely used in biomedicine or medicine (such as certified by FDA, EMA, NMPA, and so on)? In addition, PEEK may not be good alternative material in terms of price (too expensive?). It is suggested to state it in the introduction or discussion.
Response 1: As a plastic biomaterial, PEEK has been widely used in the clinical application of orthopaedic and stomatology. PEEK medical devices have been approved by regulatory bodies, such as SYNFIX Evolution Implant from Depuy Synthes, LYDESDALE PTC from Medtronic and AVS TL from Stryker. In recent years, PEEK implants developed for use in artificial joints and trauma treatments have also entered the market (doi: 10.12336/biomatertransl.2022.02.001; https://doi.org/10.1016/j.arabjc.2020.102977). (Line 147) As for the price, in the section of introduction, the cost of PEEK material on the price of medical device was state (Line 166).
Point 2. Please include the hypothesis of this experiment in the last paragraph of the introduction and state in the discussion whether the hypothesis was accepted or rejected.
Response 2: Hypothesis has been added in the context (Please see lines 175-177 and was accepted in lines 586-588 for details).
Point 3. The author mentioned that Young's modulus and yield strength of the PEEK materials used are 1.69 GPa and 85.5 MPa, respectively. Generally, Young's modulus of pure PEEK is between 3-6 GPa. Is the PEEK raw material (450G, VIC- 165 TREX, UK) modified or added with other substances to reduce its elastic modulus? In addition, the melting point of PEEK is about 340~380°C, and the nozzle temperature of the FDM 3D printer is 420°C. Will this cause carbonization of the printed PEEK?
Response 3: The pure PEEK 450 G from Victrex was used. Neither modification nor the adding of other substances was done in this study. The low elastic modulus original from the low crystallinity of the 3D printed PEEK manufactured by fused filament fabrication (FFF) technology. In previous study from the authors, the mechanical properties and of crystallinity 3D printed PEEK by FFF was carefully measured.(https://doi.org/10.1016/j.jmatprotec.2017.04.027; https://doi.org/10.1016/j.jmbbm.2021.104475) The printing temperature of PEEK of 420°C was also determined by the previous study using the same 3D PEEK 3D printer. Lower printing temperature such as 360°C or 380°C will results in weak bonding strength between printing lines. Printing temperature of 420°C was proved to be able to get parts with high strength and elastic modulus. The decomposition onset temperature of PEEK was 575°C (DOI:10.1016/j.polymdegradstab.2010.01.024) while the carbonization temperature is higher than 800°C, therefore the printing temperature of 420°C will not cause thermal decomposition or carbonization.
Reviewer 2 Report
1. The abstract section should be enhanced to include quantitative data.
2. As your abstract's final sentence, include a "take-home" message.
3. Keywords should be reordered based on alphabetical order.
4. It is encouraged not used abbreviations in the keywords section.
5. It is unclear whether the author's something new in this work. According to evaluation, several published studies by other researchers in the past adequately explain the issues you made in the present paper. Please be careful to highlight in the introduction section anything really innovative in this work.
6. In order to highlight the gaps in the literature that the most recent research aims to fill, it is crucial to review the benefits, novelty, and limitations of earlier studies in the introduction.
7. Line 117, PAEK? What is the urgency for mention PAEK if the manuscript focusses on PEEK?
8. This is a crucial element that authors must contain in the introduction and/or discussion section. Furthermore, the MDPI's suggested reverence should be applied in the manner described below to further support this description as follows:
9. Line 55, related to explanation of aseptic loosening needs supporting reference. The MDPI's suggested reverence should be adopted as follows: Jamari, J.; Ammarullah, M. I.; Santoso, G.; Sugiharto, S.; Supriyono, T.; Prakoso, A. T.; Basri, H.; van der Heide, E. Computational Contact Pressure Prediction of CoCrMo, SS 316L and Ti6Al4V Femoral Head against UHMWPE Acetabular Cup under Gait Cycle. J. Funct. Biomater. 2022, 13, 64. https://doi.org/10.3390/jfb13020064
10. Rather than relying just on the predominate text as it already exists, the authors could incorporate more illustrations as figures in the materials and methods section that illustrate the workflow of the current study.
11. Meshing strategy needs more explanation.
12. Where is boundary condition illustration? Provide it.
13. Outcomes must be compared to similar past research.
14. The limitation of the current study must be included at the end of the discussion section.
15. In the conclusion, please explain the further research.
16. The reference should be given additional literature from the recent five years for enriching it. MDPI literature is highly recommended.
17. In the whole of the manuscript, the authors sometimes made a paragraph only consisting of one or two sentences that made the explanation not clearly understood. The authors need to extend their explanation to become a more comprehensive paragraph. In one paragraph, it is recommended to consist of at least 3 sentences with 1 sentence as the main sentence and the other sentences as supporting sentences.
18. English needs to be proofread due to grammatical errors and English style, using the MDPI English editing service would be a solution.
19. Please be aware that the authors followed the MDPI format correctly; modify the current form and recheck, as well as any other problems that have been highlighted.
20. A graphical abstract is suggested to be included in the submission after peer review.
Author Response
Response to Reviewer 2:
Point 1. The abstract section should be enhanced to include quantitative data.
Response 1: Stiffness value of the Ti6Al4V and PEEK implant were added (line 34). Maximum von Mises stress values obtained in the medial neck and the distal restriction point of the implant were also added (lines 37-38) in the abstract.
Point 2. As your abstract's final sentence, include a "take-home" message.
Response 2: The final sentence (line 39) of the abstract has been modified as following: “Overall, considering the reduction in stress shielding and bone resorption in cortical bone, PEEK could be a promising material for the patient-specific femoral implants.”
Point 3. Keywords should be reordered based on alphabetical order.
Response 3: Keywords have been rearranged based on alphabetical order.
Point 4. It is encouraged not used abbreviations in the keywords section.
Response 4: Thank you for your kind suggestion. Generally, polyetheretherketone is known as PEEK in this field and there are many papers in the Polymers journal which have also used this abbreviation as their keyword. It would be helpful for the readers interest to keep this abbreviation as PEEK.
Point 5. It is unclear whether the author's something new in this work. According to evaluation, several published studies by other researchers in the past adequately explain the issues you made in the present paper. Please be careful to highlight in the introduction section anything really innovative in this work.
Response 5: This is the first study which evaluates the stress shielding effect, bone resorption and mechanical behaviour of hip stem fabricated via additive manufacturing technology. In this study, PEEK has been used to manufacture the whole stem and not just as surface coating material similar to previous studies. This is now clearly mentioned in the context (please reference to lines 164-166 for details).
Point 6. In order to highlight the gaps in the literature that the most recent research aims to fill, it is crucial to review the benefits, novelty, and limitations of earlier studies in the introduction.
Response 6: Introduction has been modified with more literature stating the benefits, novelty, and limitations of earlier studies (lines 139-142). Table 1 has been added in the introduction.
Point 7. Line 117, PAEK? What is the urgency for mention PAEK if the manuscript focusses on PEEK? This is a crucial element that authors must contain in the introduction and/or discussion section. Furthermore, the MDPI's suggested reverence should be applied in the manner described below to further support this description as follows:
Response 7: PEAK is mentioned to briefly introduce that PEEK is from PEAK polymer family. Some studies used both PEAK and PEEK as coating materials for hip stems design which is addressed in line 158.
Point 8. Line 55, related to explanation of aseptic loosening needs supporting reference. The MDPI's suggested reverence should be adopted as follows: Jamari, J.; Ammarullah, M. I.; Santoso, G.; Sugiharto, S.; Supriyono, T.; Prakoso, A. T.; Basri, H.; van der Heide, E. Computational Contact Pressure Prediction of CoCrMo, SS 316L and Ti6Al4V Femoral Head against UHMWPE Acetabular Cup under Gait Cycle. J. Funct. Biomater. 2022, 13, 64. https://doi.org/10.3390/jfb13020064
Response 8: Relevant reference has been added in line 61 as reference 4.
Point 9. Rather than relying just on the predominate text as it already exists, the authors could incorporate more illustrations as figures in the materials and methods section that illustrate the workflow of the current study.
Response 9: Figure 1 has been added on page 5 which shows the flow chart of general strategies used in this study.
Point 10. Meshing strategy needs more explanation.
Response 10: Thanks for your comment. Mesh type (10-node quadratic tetrahedron -C3D10) has been described in line 336. Mesh convergence was performed with element sizes ranging from 0.1 mm to 3 mm (lines 339-340). It was demonstrated that the results were converged within 5% with an element size of 0.65 mm (lines 341-342). Manual mesh seeding was carried out to increase the number of elements in the Gruen zone with element size of 0.5 mm (lines 344-345). Please reference to relevant sections in the manuscript for details.
Point 11. Where is boundary condition illustration? Provide it.
Response 11: Loading and boundary conditions (Figure 6) has been added on page 11.
Point 12. Outcomes must be compared to similar past research.
Response 12: Results of this study has been compared to previous studies, please reference to lines 586-593 for details.
Point 13. The limitation of the current study must be included at the end of the discussion section.
Response 13: Limitations of the current study have been added in section 3.6 on page 21 lines 663-672.
Point 14. In the conclusion, please explain the further research.
Response 14: Future work has been added in the conclusion, please reference to lines 699-704 for details.
Point 15. The reference should be given additional literature from the recent five years for enriching it. MDPI literature is highly recommended.
Response 15: Eight more recent studies were added in the introduction, 4 of which were from MDPI journal.
Point 16. In the whole of the manuscript, the authors sometimes made a paragraph only consisting of one or two sentences that made the explanation not clearly understood. The authors need to extend their explanation to become a more comprehensive paragraph. In one paragraph, it is recommended to consist of at least 3 sentences with 1 sentence as the main sentence and the other sentences as supporting sentences.
Response 16: This has been corrected.
Point 17. English needs to be proofread due to grammatical errors and English style, using the MDPI English editing service would be a solution.
Response 17: Thanks for your recommendation. The revised manuscript has been checked and edited by a native English-speaking colleague.
Point 18. Please be aware that the authors followed the MDPI format correctly; modify the current form and recheck, as well as any other problems that have been highlighted.
Response 18: Authors have tried their best to follow the MDPI format and suggested modifications have been performed accordingly.
Point 19. A graphical abstract is suggested to be included in the submission after peer review.
Response 19: A graphical abstract has been prepared and is submitted as JPG file in the revised manuscript.
Round 2
Reviewer 2 Report
My recommendation is to accept the manuscript in the present form.